# Balancing Waste and Nutrient Flows Between Urban Agglomerations and Rural Ecosystems: Biochar for Improving Crop Growth and Urban Air Quality in The Mediterranean Region

**Anastasia Zabaniotou ***  **and Katerina Stamou**

Biomass Group, Department of Chemical Engineering, Engineering School, Aristotle University of Thessaloniki, 54636 Thessaloniki, Greece; katerina_stamou@outlook.com

\* Correspondence: azampani@auth.gr; Tel.: +3069-4599-0604

**Abstract:** Mediterranean ecosystems are threatened by water and nutrient scarcity and continuous loss of soil organic carbon. Urban agglomerations and rural ecosystems in the Mediterranean region and globally are interlinked through the flows of resources/nutrients and wastes. Contributing to balancing these cycles, the present study advocates standardized biochar as a soil amendment, produced from Mediterranean suitable biowaste, for closing the nutrient loop in agriculture, with parallel greenhouse gas reduction, enhancing air quality in urban agglomerations, mitigating climate change. The study's scope is the contextualization of pyrolytic conditions and biowaste type effects on the yield and properties of biochar and to shed light on biochar's role in soil fertility and climate change mitigation. Mediterranean-type suitable feedstocks (biowaste) to produce biochar, in accordance with biomass feedstocks approved for use in producing biochar by the European Biochar Certificate, are screened. Data form large-scale and long-period field experiments are considered. The findings advocate the following: (a) pyrolytic biochar application in soils contributes to the retention of important nutrients for agricultural production, thereby reducing the use of fertilizers; (b) pyrolysis does not release carbon dioxide to the atmosphere, contributing positively to the balance of carbon dioxide emissions to the atmosphere, with carbon uptake by plant photosynthesis; (c) biochar stores carbon in soils, counterbalancing the effect of climate change by sequestering carbon; (d) there is an imperative need to identify the suitable feedstock for the production of sustainable and safe biochar from a range of biowaste, according to the European Biochar Certificate, for safe crop production.

**Keywords:** biochar; pyrolysis; waste; soil amendment; air quality; climate change; Mediterranean

---

## 1. Introduction

The increasing rates of human population in urban agglomerations of Mediterranean regions and the unbalanced urban-rural interlinks create ecological cross-boundary challenges and negative resource/climate change outcomes. Solutions for better balancing the unsustainable urban-rural flows, without endangering the rural ecosystems and human health, are in the focus of the scientific community. Mediterranean ecosystems' sustainable management is a fundamental element of the environmental sustainability of the region, biosphere, hydrosphere, geosphere, atmosphere, and their interrelations [1].

Biochar has attracted much attention globally, due to its promising role in enhancing crop growth, by serving as a soil amendment, and in enhancing air quality, boosting sustainable agriculture in rural areas and health in urban agglomerations. Among the agricultural conservation practices that mitigate some of the adverse impacts of land use intensification, biochar is an effective input for

sustainable agriculture, as it can efficiently sequester large amounts of carbon in soil over the long-run, thus improving soil fertility, crop productivity, and global warming mitigation [2]. Biochar is proposed to be used as a soil enhancer for a sustainable agricultural system, because it can enhance water and nutrient retention in the soil, due to its porous nature and the contained oxygen functional groups and aromatic compounds on its surface [3]. Application of biochar to soil may provide agronomic, environmental, and economic benefits, as it plays a significant role in the alteration of nutrient dynamics, soil contaminants, as well as microbial functions [4].

Although biochar is widely suggested as a soil amendment to improve soil physical properties for crop production, it is difficult to compare results from different studies due to the heterogeneity of experimental procedures in terms of experimental conditions, feedstock type, biochar characteristics, and soil properties [5]. Biochar as remediation of poor soils (low fertility soils) by coating with organic materials and accelerating the composting process for enhanced crop nutrient supply has also been suggested by other researchers [6,7]. Biochar is also strongly recommended as one of the best management practices to meet the challenges of upland agriculture [8].

Large-scale field experiments were performed and published recently with biochar, under pilot and demonstration projects. A Special Issue entitled "Biochar as Soil Amendment: Impact on Soil Properties and Sustainable Resource Management" of the journal of Agronomy that was published in 2019 includes many large-scale field applications of biochar studies, showing the growing interest in biochar production from any kind of biowaste [9]. Some of the published papers are very relevant to Mediterranean countries. For example, the effects of biochar on wheat productivity is of interest for Mediterranean countries, since they are wheat producers [10]. The same goes for the results of a study that proved the incorporation of a combination of biochar and rice straw in paddy soil to increase the yield of rice grains, compared to unamended soil. This also is relevant because of rice production in many Mediterranean countries, European and African [11]. The valorization of vineyard by-products to obtain biochar suitable for nursery grapevine is very relevant due to wine production in Mediterranean countries [12]. Studies on sewage sludge pyrolysis for biochar production [13] are relevant for all Mediterranean agglomerations, and the study of sandy soils is relevant for the soils of African Mediterranean countries [14].

Besides biochar's benefits, limitations of its application in agriculture exist, and it must be taken into consideration prior to biochar's utilization [15]. The application of unsuitable biochar can negatively affect environmental quality and human health due to reduction in plant nutrient uptake and harmful compounds such as polycyclic aromatic hydrocarbons (PAHs), polychlorinated dibenzodioxins (PCDD), and dibenzofurans (DF) [16].

Besides soil amendment, biochar has emerged as a promising material for adsorbing and thus decreasing the bioavailability of pesticides in polluted soils. Researchers reported studies on soil remediation using biochar as an environmentally-friendly amendment to counteract the presence of pesticides [17]. The contribution to anthropogenic climate change mitigation, enhancing climate stability, is also attributed to biochar. A recently published paper systematically reviewed studies on life cycle assessment (LCA) to assess the environmental impact of biochar on soils, and it concluded that carbon sequestration is the main beneficial process of biochar [18].

The recent interest in using biomass and biowaste for char production drove many researchers to address in their research some of the energy and environmental problems we are facing today, such as $CO_2$ emissions, the energy crisis, and environmental pollution. These studies advance engineered biochar technology and demonstrate insights on new directions of research and innovation [19]. Lignocellulosic biomass as a precursor of biochar via thermal decomposition in limited oxygen conditions is suggested as a sustainable adsorbent of heavy metals [20]. Other researchers explored the environmental and energy benefits of biomass residues, including crop residues and agricultural waste, to produce biochar to be used as fuel [21].

Many processes are reported to be suitable for biochar production. A recent review study advocated that hydrothermal liquefaction is good because it produces a biochar suitable to retain

essential functional groups compared to other biochar [22]. Another review study gave great emphasis to gasification aiming at balancing syngas and biochar production for proposing economically- and environmentally-feasible gasification systems. The study also qualitatively evaluated biochar sustainability via life cycle assessment (LCA) [23]. Pyrolysis is reported as a process to produce biochar able to use various feedstocks under different process conditions. It is proposed to be used for agricultural and environmental practices in real-world applications [24] within circular economy context.

The best thermochemical biochar production method is slow pyrolysis. Pyrolysis temperature, heating rate, residence time, and the type of waste have a significant effect on both the properties and yield of biochar [3,25]. A pyrolysis-biochar system for Mediterranean olive farm residues in symbiosis with a two-phase olive mill was proposed as a circular economy scenario for the olive oil production supply chain's waste management, with simultaneous carbon sequestration and soil improvement coupled with bio-energy generation by a Mediterranean research team. The study advocated that that pyrolysis of agri-residues targeting biochar could fulfill the aim of closing the loops in agriculture and circular economy objectives [25]. A more recent paper reviewed potential opportunities for food waste pyrolysis to biochar products and found that more research and development work needs to be conducted for food waste-to-biochar options [26].

Mediterranean agri-ecosystems are threatened by water scarcity and suffer from severe losses of soil organic carbon (SOC), leading to a high risk of land degradation. Recycling strategies to close the loops among agriculture, agro-industrial sector, urban waste management, and soil conservation are of special interest in the current context of climate change mitigation and the EU's circular economy strategy. Pyrolysis represents a recycling option to produce biochar, a carbonaceous product with a wide range of environmental and agronomic applications [27].

Despite the number of detailed studies describing the effects of biochar, there is a lack of knowledge concerning the real carbon sequestration potential of biochar in amended soils under the Mediterranean climate, and also, the urban-rural flows of the region remain unstudied.

The present review deals with an overview of the currently available knowledge on the above subject supported by the authors' own research results and knowledge. Four categories of biomass are selected for generating biochar at varied pyrolysis conditions, governing the biochar properties. This categorization is essential to rank suitable feedstocks to produce biochar in Mediterranean countries, assuming its safe use in crops and food production. Varied properties of feedstock materials and the resultant biochar produced due to different production processes influencing their chemical, physical, and structural properties are summarized. Appropriate feedstocks are the limitation for the definition of the four categories of waste considered in this review, comprised of (a) agricultural residues, (b) lignocellulosic residues, (c) animal wastes, and (d) sludge not containing heavy metals. Sludge from municipal sewage, pulp and paper mill effluent, and slaughterhouse sludge have high potential for toxicity, due to high contents of heavy metals (HMs) such as Cu, Cr, Pb, Ni, Cd, and Zn [24]; therefore, they are excluded for the purpose of this review. The resultant properties of biochar are vital to appreciate the functionality of biochar in the soil and the potential to control GHG emissions.

The aim of this review article is threefold:

i.　The collection of scientific arguments for advocating the valorization of Mediterranean biowaste via pyrolysis as an upcycling method to produce biochar.

ii.　Supporting biochar's safe use by re-grouping European and International standards and methods.

iii.　Advocating biochar's agricultural and environmental benefits, such as:

　(a)　nutrient retention,
　(b)　closing cycles between urban and rural ecosystems
　(c)　producing better air quality in cities by sequestrating carbon in soils.

Urban agglomerations and rural ecosystems in the Mediterranean countries are interlinked through the flows of resources/nutrients and wastes, which are unbalanced and mostly unstudied. The scientific objectives of this review study are:

1.  The performance of a meta-analysis of selected experimental data from papers of the international literature concerning the biowaste pyrolysis process, as a convenient thermochemical method whereby biowaste is efficiently converted into biochar.
2.  The performance of a parametric meta-analysis of engineered pyrolytic biochar characteristics
3.  The assessment of current development work and evaluation of potential opportunities for available biowaste in the European Mediterranean countries (Greece, Italy, France, Spain, Portugal, etc.), but also for other non-European Mediterranean countries (Turkey, Egypt, etc.) to produce pyrolytic biochar for complementary sustainable agriculture practices and climate change mitigation alternatives.

## 2. Methodology

In this study, a critical literature review was performed following the methodology proposed by Thürer et al. (2018) [28], for searching for and analyzing internationally published articles. The aim was to retrieve and select the appropriate publications relevant to the present research topic and corresponding to experimentations and results that were relevant to Mediterranean types of biowaste as precursors of biochar production and soil chemical properties.

The bibliographic databases used for sourcing the articles were Web of Science (2009–2019), ScienceDirect (2009–2020), Google Scholar (2009–2019), MDPI (2009–2019), and open access publications. It is recognized that there is an extensive literature in the form of books, but it was not possible to have access to all relevant books for a systematic review; however, we used some. In order to keep the number of articles reasonable and to ensure the quality of the sources, the search was further restricted to peer-reviewed articles. To keep results to a manageable number, the search was restricted to the title, abstract, and keywords of papers. Document type was limited to "articles" and reviews. It was decided that the final sample would be limited to papers that had been cited. There was a restriction on the year of publication of the journals considered, for the decade of 2009-2019, because most of the scientific publishing on biochar appeared after 2009.

During the revision of the manuscript to its updated version, thirteen publications of the year 2020 were also searched and cited; however, these newly published articles were accepted even without citations, since they all are from the year 2020, but they are not given in the Figures 1 and 2.

The search was carried out with the following keywords "biochar" AND "properties", "biochar" AND "soil amendment", "biochar" AND "greenhouse gases' emissions/climate change". Various online magazines such as "Science of the Total Environment", "Agriculture, Ecosystems and Environment", and "Biology and Fertility of Soils" and ''Agronomy" were searched.

The first search resulted in 4386 publications (on the use of biochar as a soil improver), made by using the keywords "biochar AND soil amendment". Three-thousand six-hundred twenty-six (3626) publications were excluded, as they were not directly relevant to this topic of study. Thus, seven-hundred sixty (760) related articles remained for furthermore detailed screening. From this set, one-hundred twenty-five (120) publications were used as the baseline for this review, containing necessary information and data for this study.

The search addressed to biowaste as the precursor material of biochar was comprised of 4 categories of biowaste, typical in the Mediterranean region:

*   agricultural and agri-food waste,
*   lignocellulosic waste,
*   animal waste,
*   sewage sludge.

Therefore, the final selection of papers was based on experiments with biochar issued from biowaste that exists in Mediterranean countries. Biochar from toxic solid waste was excluded in this study because this up-cycled biochar may have potential risks by secondary infection of crops [4].

Table 1 presents the results of the screening process of the papers selected from the international literature by using the words: "biochar AND soil amendment".

**Table 1.** Screening of articles on the theme "biochar application for enhancing soil fertility".

| Screening Process | Number of Remained Articles in the Sample |
|---|:---:|
| 1st Screening (1st sample) | |
| (research and review articles) | **4386** |
| Review articles only (within the 1st sample) | 496 |
| Research articles | 3890 |
| 2nd Screening (2nd sample) | |
| (research and review articles) | **760** |
| Publications with Mediterranean-type waste | 41 |
| Other | 719 |
| 3rd Screening (studied sample with cited references) | |
| Papers relevant to biochar for soil amendment | **125** |
| Papers exclusively referring to biochar as a soil improver | 90 |
| Others with combined relevance | 35 |
| Total number of citations in this paper: 131 (125 from Table 1+6 from Table 2) | |

The following Figure 1 presents a statistical analysis of the published articles regarding biochar as a soil improver versus the year the of publication.

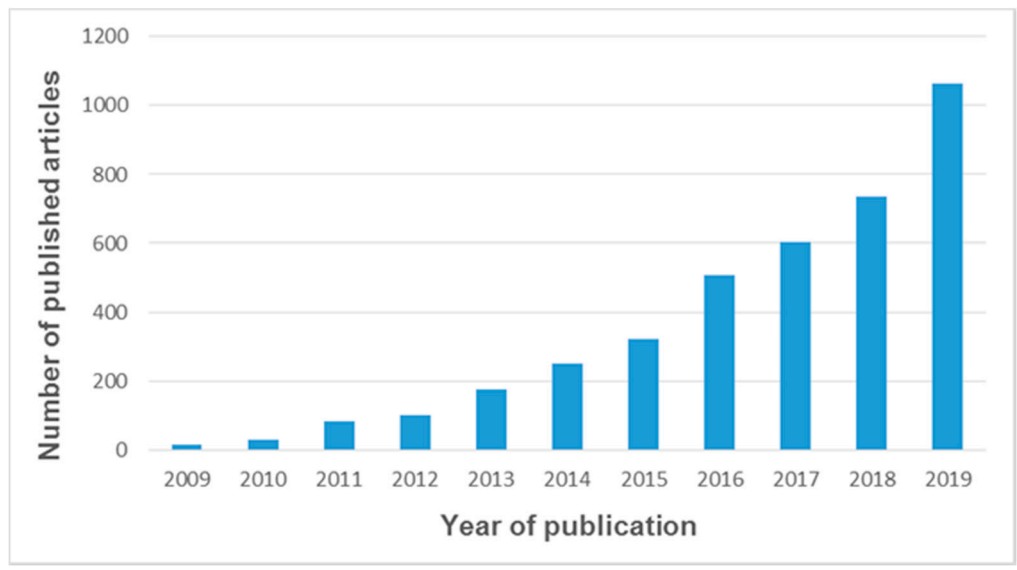

**Figure 1.** Statistical analysis of articles related to the topic of "biochar as soil enhancer".

From Figure 1, it is observed that at the end of the decade 2009–2019, a growing interest in biochar appeared, evidenced by the increased research activity on the application of biochar in soil and the potential benefits it can offer to soil quality and fertility.

A second search for the use of biochar as an effective agent contributing to climate change mitigation via the greenhouse gas emissions decrease through the use of the pyrolysis process was performed by using the keywords: "biochar AND greenhouse gases emissions (GHGs)/mitigation" (Table 2). This resulted in 3266 publications. Two-thousand nine-hundred sixty-four (2964) papers were excluded, as not being directly related to the specific research topic. Thus, three-hundred two

(302) relevant articles for further research were kept. Of this total, forty-one (41) publications were used as the baseline in this study, after a more detailed screening having as the criterion the type of the precursor waste for biochar production (Mediterranean relevance). Only papers exploring biochar issued from biowaste that was Mediterranean relevant were considered. The others were omitted.

**Table 2.** Screening of articles on the topic "biochar application for GHGs mitigation".

| Screening Process | Number of Articles |
|---|---|
| 1st Screening (1st sample) | |
| (research and review articles) | **3226** |
| Review articles only (within the 1st sample) | 569 |
| Research articles | 2657 |
| 2nd Screening (2nd sample) | |
| (research and review articles) | **302** |
| Publications with Mediterranean-type waste relevance only | 46 |
| Others with combined relevance | 256 |
| 3rd Screening (studied sample with cited references) | |
| Papers relevant to biochar for climate change mitigation | 41 |
| Papers exclusively referred to biochar for soil mitigation | **6** |
| Other | 35 |
| Total number of citations in this paper: 131 (125 from Table 1+6 from Table 2) | |

Figure 2 presents a statistical analysis of the published articles regarding "biochar as an agent of minimizing greenhouse gas emissions/climate change", per year, for the period from 2009-2019. It is observed that there was also a growing interest in the topic, evidenced by the exponential trend of research activity. Compared with the results of Figure 1, the trend on the second topic research activity was almost like the first topic, with the second having higher absolute numbers of papers published.

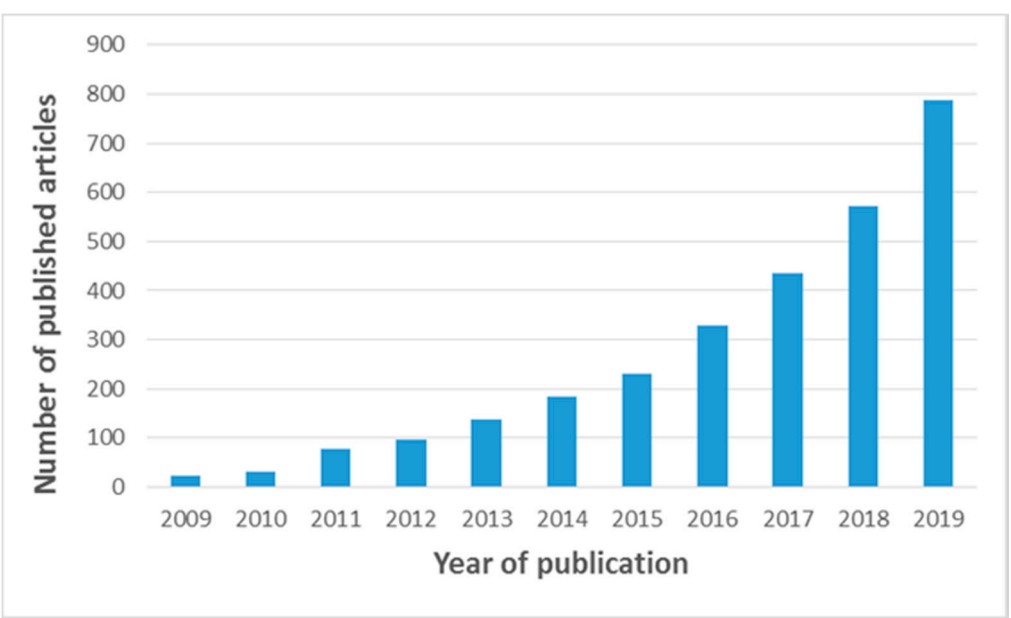

**Figure 2.** Statistical analysis of articles related to the topic of "biochar application for greenhouse gases emissions (GHG) mitigation".

　　From the bibliographic search and as depicted in Figures 1 and 2, it is obvious that research work on the production and application of the biochar has very dynamically increased internationally since 2011, and the application of biochar in soils has been widely discussed, mainly referring to sequestration of carbon and improvement of soil's chemical and physical properties for better plant growth, with parallel waste management. This increasing interest goes together with the evolving circular economy concept, and of course with the IPCC directive on climate change and carbon sequestration.

　　Screening all the relevant articles, papers containing useful experimental data about biochar related to Mediterranean waste were selected and are given in Tables 3–5. These tables contain the properties of biochar produced by 4 main categories of Mediterranean-type biowastes (agricultural residues, animal wastes, lignocellulosic wastes, and sludges).

**Table 3.** Properties of biochar derived from various wastes at various pyrolysis process parameters.

| Pyrolysis Feedstock | Pyrolysis Temperature (°C) | Pyrolysis Heating Rate (°C/min) | Biochar Yield (wt.%) | Biochar Volatile Matter (wt.%) | Biochar Fixed Matter (wt.%) | Biochar Ash (wt.%) | Biochar Surface Area (m$^2$/g) | Reference |
|---|---|---|---|---|---|---|---|---|
| **Agricultural waste** | | | | | | | | |
| Corn stover | 450 | | 15.0 | 12.7 | 28.7 | 58.0 | 1.2 | [29] |
| | 500 | | 17.0 | - | - | 32.8 | 3.1 | |
| Wheat straw | 300 | | 46.96 | | | 14.39 | 1.9 | [30] |
| | 400 | | 35.76 | | | 18.28 | 2.6 | |
| | 500 | | 32.49 | | | 22.39 | 3.3 | |
| | 600 | | 31.55 | | | 21.82 | 4.5 | |
| Rice straw | 400 | | 39.3 | 22.42 | | | 46.60 | [31] |
| | 500 | | 32.6 | 12.80 | | | 59.91 | |
| | 600 | | 23.4 | 8.36 | | | 129.00 | |
| Rice husk | 400 | | 48.6 | 22.00 | | | 193.70 | [31] |
| | 500 | | 42.4 | 10.56 | | | 103.17 | |
| | 600 | | 37.3 | 6.02 | | | 288.58 | |
| Cotton husk | 400 | | 38 | | | | 0.2 | [31] |
| | 600 | | 33 | | | | 1.9 | |
| Corn cob | 300 | | | 43.60 | 49.10 | 4.90 | | [32] |
| | 500 | | | 8.60 | 81.60 | 8.20 | | |
| | 600 | | | 7.20 | 82.40 | 8.70 | | |
| **Animal wastes** | | | | | | | | |
| Poultry manure | 500 | 7.0 | 72.0 | 7.3 | 68.6 | 24.0 | 5.8 | [33] |
| | 700 | 7.0 | 47.0 | 4.1 | 69.6 | 24.2 | 6.6 | |
| Poultry litter | 300 | | 60.13 | | | | 1.75 | [34] |
| | 400 | | 51.52 | | | | 5.65 | |
| | 500 | | 47.57 | | | | 18.12 | |
| | 600 | | 45.71 | | | | 25.33 | |
| **Lignocellulosic wastes** | | | | | | | | |
| Wood sawdust | 300 | | 4.38 | | | 1.28 | | [30] |
| | 500 | | 4.56 | | | 3.76 | | |
| | 700 | | 5.73 | | | 7.53 | | |
| Apple tree branch | 400 | | 28.3 | 32.36 | | | 11.90 | [32] |
| | 500 | | 16.7 | 18.27 | | | 58.60 | |
| | 600 | | 16.6 | 11.07 | | | 208.69 | |
| Oak tree | 300 | | 35.8 | 32.06 | | | 5.60 | [32] |
| | 400 | | 28.6 | 19.42 | | | 103.17 | |
| | 500 | | 22.0 | 12.30 | | | 288.58 | |
| | 600 | | 20.0 | 8.28 | | | 335.61 | |
| Eucalyptus sawdust (E. saligna) | 400 | | 41 | | | | 0.3 | |
| | 600 | | 33 | | | | 132.0 | |
| **Sewage sludge** | | | | | | | | |
| Sewage sludge | 300 | 92.19 | | | | 68.62 | 81.66 | [35] |
| | 400 | 81.66 | | | | 70.14 | 74.73 | [36] |
| | 500 | 67.80 | | | | 79.00 | 67.8 | |
| | 600 | 65.12 | | | | 85.75 | 65.12 | |

**Table 4.** Elemental composition, pH and cation exchange capacity (CEC) of biochar produced from various wastes at different pyrolysis temperature.

| Pyrolytic Biochar Feedstock | Pyrolysis T (°C) | pH | C (wt.%) | H (wt.%) | O (wt.%) | N (wt.%) | CEC (kmole/kg) | Reference |
|---|---|---|---|---|---|---|---|---|
| **Agricultural residues** | | | | | | | | |
| Corn stover | 450 | - | 33.2 | 1.40 | 8.60 | 0.81 | | [29] |
| | 500 | 7.2 | 57.29 | 2.86 | 5.45 | 1.47 | | |
| Wheat straw | 300 | 7.98 | 61.48 | 2.73 | 19.61 | 1.40 | | |
| | 400 | 9.06 | 64.18 | 1.78 | 13.93 | 1.36 | | |
| | 500 | 10.37 | 67.39 | 1.01 | 7.35 | 1.38 | | |
| | 600 | 10.83 | 65.34 | 0.52 | 10.77 | 1.10 | | [30] |
| Rice Straw | 300 | | 73.59 | 4.46 | | 0.69 | | |
| | 500 | | 72.45 | 3.08 | | 0.46 | | |
| | 700 | | 60.27 | 1.46 | | 0.38 | | |
| Rice straw | 400 | 8.62 | 49.92 | 2.80 | 12.02 | 1.22 | | |
| | 500 | 9.82 | 37.48 | 0.93 | 8.64 | 0.61 | | |
| | 600 | 10.19 | 33.78 | 0.60 | 13.68 | 0.41 | | |
| | 700 | 10.39 | 36.26 | 0.51 | 17.38 | 0.34 | | |
| | 800 | 10.47 | 29.17 | 0.25 | 3.71 | 0.25 | | [31] |
| Rice husk | 400 | 6.84 | 44.59 | 2.50 | 16.32 | 0.69 | | |
| | 500 | 8.99 | 45.15 | 1.27 | 7.12 | 0.47 | | |
| | 600 | 9.41 | 40.35 | 0.85 | 9.23 | 0.37 | | |
| | 700 | 9.52 | 38.81 | 0.46 | 12.69 | 0.26 | | |
| | 800 | 9.62 | 40.41 | 0.28 | 2.69 | 0.22 | | |
| Cotton husk | 400 | 10.0 | 69.8 | | | 2.0 | 49.0 | [37] |
| | 600 | 10.0 | 65.6 | | | 1.8 | 56.8 | |
| Corn cob | 300 | | 67.21 | 4.49 | 27.63 | 0.67 | | |
| | 500 | | 83.27 | 3.33 | 12.62 | 0.78 | | |
| | 600 | | 84.31 | 2.41 | 12.52 | 0.76 | | [32] |
| Maize straw | 300 | | 57.40 | 6.64 | 34.20 | 1.59 | | |
| | 500 | | 80.70 | 3.23 | 14.10 | 1.71 | | |
| **Animal wastes** | | | | | | | | |
| Poultry | 500 | 11.0 | 51.56 | 1.87 | 40.32 | 5.50 | | [33] |
| manure | 700 | 10.7 | 56.09 | 1.52 | 37.19 | 4.16 | | |
| Swine manure | 400 | 9.2 | 49.6 | | | 2.7 | 28.9 | [37] |
| | 600 | 10.7 | 47.0 | | | 1.8 | 45.4 | |
| Poultry litter | 300 | 6.29 | 25.28 | | | | | |
| | 400 | 9.54 | 26.96 | | | | | [34] |
| | 500 | 9.99 | 28.91 | | | | | |
| | 600 | 10.06 | 29.01 | | | | | |
| Bull manure | 300 | 8.2 | 60.6 | | | 1.3 | | |
| | 600 | 9.5 | 76.0 | | | 0.8 | | |
| Dairy manure | 350 | 9.2 | 55.8 | | | 2.60 | | [38] |
| | 400 | 9.22 | 57.7 | | | 0.242 | | |
| | 600 | 9.94 | 59.4 | | | 0.225 | | |
| | 700 | 9.9 | 56.7 | | | 1.51 | | |
| **Lignocellulosic waste** | | | | | | | | |
| Wood sawdust | 300 | | 76.45 | 2.67 | | 0.65 | | |
| | 500 | | 84.32 | 1.83 | | 0.54 | | [30] |
| | 700 | | 89.92 | 1.36 | | 0.41 | | |
| Apple tree branch | 400 | 7.02 | 70.18 | 4.13 | 20.56 | 0.76 | | |
| | 500 | 9.64 | 79.12 | 2.65 | 11.98 | 0.34 | | |
| | 600 | 10.04 | 81.46 | 1.96 | 13.63 | 0.46 | | |
| | 700 | 10.03 | 82.26 | 1.21 | 16.34 | 0.41 | | |
| | 800 | 10.02 | 84.84 | 0.60 | 5.81 | 0.34 | | [31] |
| Oak tree | 300 | 6.84 | 44.59 | | | | | |
| | 400 | 8.99 | 45.15 | | | | | |
| | 500 | 8.85 | 81.22 | | | | | |
| | 600 | 9.54 | 83.22 | | | | | |
| Eucalyptus sawdust (E. saligna) | 400 | 7.7 | 78.5 | | | 0.7 | 3.7 | |
| | 600 | 9.6 | 84.0 | | | 0.8 | 19.8 | [37] |
| **Sludge** | | | | | | | | |
| Sewage sludge | 200 | 6.54 | 17.09 | 2.09 | 10.01 | 2.19 | | |
| | 300 | 7.20 | 19.72 | 1.79 | 5.76 | 2.59 | | [35] |
| | 500 | 8.70 | 15.26 | 0.73 | 3.28 | 1.73 | | [36] |
| | 700 | 11.15 | 11.33 | 0.31 | 1.90 | 0.71 | | |

**Table 5.** Large-scale open field applications of biochar derived from various feedstocks and impact on crop yield results.

| Pyrolysis Feedstock | Biochar Application rate (t/ha) | Soil Type | Crop | Crop yield Increase/Decrease (wt.%) | Reference |
|---|---|---|---|---|---|
| **Agricultural Waste** | | | | | |
| Green wastes | 6.75 <br> 13.5 <br> 40.5 | Potting mixture | Cucumber | +99 <br> +81 <br> 30 | [32] |
| | 10 <br> 50 <br> 100 | Alfisol | Radish | -30 <br> +91 <br> +130 | [39] |
| Maize straw | 0.45 | Entic Hydroagric Anthrosol | Rice | +10.46 | [40] |
| | 2.4 | Sandy loam | | +6 | [41] |
| Wheat Straw | - | Acid soil | Wheat | +19.6 | [35] |
| | | | Millet straw | +60.6 | |
| Corn straw | - | Sandy soil | Cotton | +9.2 | [42] |
| | 20 <br> 10 <br> 5 | Inceptisol | | +21 <br> +18 <br> +9 | |
| | - | Saline soil | Wheat | +27.7 | [43] |
| Wheat straw | 1 <br> 5 <br> 10 | Acid Ferrasol | Rice | +19 <br> +79 <br> +51 | [44] |
| | 10 <br> 20 <br> 40 | | | +28 <br> +9 <br> +22 | [45] |
| | 12 | Slightly alkaline sandy loam | | Neutral | [46] |
| | - | Acid soil | Sunflower | +50 | [47] |
| | | Saline soil | Wheat | +38 | [48] |
| | | | Maize | +200 | [49] |
| | | Waterlogged paddy | Wheat | +37.6 | [50] |
| | 2.5 <br> 5 <br> 10 <br> 20 <br> 30 <br> 40 | Sandy loam soil | Rapeseed | +22 <br> +22 <br> +43 <br> +37 <br> +53 <br> +61 | [51] |
| Wheat straw | 10 <br> 20 <br> 40 | Hydroagric Stagnic Anthrosol | Rice | +27.63 <br> +9.2 <br> +22.39 | [49] |
| | 10–40 | Fine loamy Gleysols | | Neutral | [49] |
| Rice straw | 5 | Alkaline sandy loam Inceptisol | Rice | +24.3 | [52] |
| | | Acidic sandy loam Alfisol | | +31.3 | |
| | 10.5 | Gley paddy | | +10 | [51] |
| | 2.25 | Silt loam | | +7 | [50] |
| | 4.5 <br> 9 | Gleyi-stagnic | Rice-wheat | +5.88 <br> +14.8 | [35] |
| **Animal Waste** | | | | | |
| Poultry manure | 30 <br> 60 <br> 12 | Acidic silty | Wheat | +28.2 <br> +28.6 <br> +38.0 | [53] |
| Poultry litter | 1 <br> 5 <br> 10 | Acidic Aeronosol | Rice | Neutral <br> Neutral <br> -21 | [44] |
| | - | Acid soil | Wheat | +89 | [44] |
| Cow manure | - | Sandy soil | Maize | +150 | [54] |

**Table 5.** *Cont.*

| Pyrolysis Feedstock | Biochar Application rate (t/ha) | Soil Type | Crop | Crop yield Increase/Decrease (wt.%) | Reference |
|---|---|---|---|---|---|
| **Lignocellulosic waste** | | | | | |
| Eucalyptus | 90<br>60 | Neutral clay loam Oxisol | Bean | +46<br>+39 | [55] |
|  | 6.75<br>13.5<br>40.5 | Potting mixture | Cucumber | +55<br>+61<br>+89 | [32] |
| Wood | - | Sandy soil | Rice | +20 | [54] |
|  |  |  | Soybean grain | +100 | [56] |
|  | 20 | Clayey | Maize | +143 | [57] |
| Hardwood | 19<br>38<br>58 | MidwesterMollisols | Maize | +10<br>+17<br>+48 | [58] |
| **Sludge** | | | | | |
| Sewage sludge |  | Pot trial | Tomato | +25<br>+34 | [59] |
|  | 6<br>9<br>12 | Sandy loam | Mung bean | +143.34<br>+180.78<br>+164.50 | [60] |
|  | 25<br>50<br>100 |  | Cucumber | +23.28<br>+43.69<br>+61.32 | [61] |

## 3. About Biochar

Biochar is a carbon-rich solid material, produced by the thermal decomposition of various waste organic matter flows at relatively low temperature, mainly in thermochemical processes with/without air presence.

### 3.1. Biochar Production

Pyrolysis as the biochar production process was researched, by focusing on slow pyrolysis because it was evidenced as the process to produce higher yields of biochar and the most environmentally-friendly technology compared with combustion and gasification [3].

Slow pyrolysis is characterized by low heating rates (°C/min) and high residence times (h). The most common reactors used are rotary kilns and screw pyrolyzers. The appropriate cracking temperature range is 300–800 °C with the most common residence time ~1–2 h [62]. It is usually carried out at atmospheric pressure, with the heat provided by the partial combustion of external energy sources, or by recycling the pyrolysis gas for energy optimization. The slow pyrolysis process favors the production of biochar with yields reaching almost 50 wt.% [63].

The slow pyrolysis process is a preferable process due to:

(a) High yield of biochar production.
(b) Low harmful emissions of SOx and NOx release, being an environmentally-friendly technology for biochar production [28].
(c) Its flexibility in handling different types of feedstocks (wastes), under various operating conditions, which makes it possible to produce designed characteristics of biochar [2].

### 3.2. Biochar Properties

Biochar shows characteristics different than its biogenic precursor feedstocks. These characteristics depend on pyrolysis conditions, including temperature, heating rate, pressure, and residence time [2,63]. They are depicted in Tables 3 and 4.

The main physical properties of biochar are (Table 3):

- Porous structure, volume, and the size of pores
- Density

- Surface area
- Water holding capacity

The chemical properties of biochar are (Table 4):

- pH
- Ash content
- Electrical conductivity
- Cation exchange capacity (CEC)
- Fixed carbon and volatile matter
- Elemental composition (C, N, H, O).
- Content of metals, some of which are heavy according to the type of feedstock (P, S, K, Ca, Mg, Fe, Cu, Zn, and Mn)

### 3.2.1. Porous Structure

The pore size of biochar varies depending on the precursor material and pyrolysis temperature. It usually ranges from nano (<0.9 nm), micro (<2 nm), to macro (>50 nm) [64]. With time, biochar's absorption capacity decreases due to pores' destruction or deactivation [64].

### 3.2.2. pH

An increasing pH value results in the increasing of biochar' alkalinity. The pH value of biochar is the most significant property for agricultural applications as soil amendments. Extreme pH values of biochar are not appropriate for their application in soil. Increasing pyrolysis temperature leads to increasing pH values of the produced biochar (Table 4) [3,65].

### 3.2.3. Density

With increasing pyrolysis temperature, gases devolatilize from the solid biomass structure and form a porous structure. The bulk density considers the volume specific weight of a bulk material in a heap or pile and includes both the pores in the solid structure, as well as the voids between different particles of the bulk [65]. Bulk density is defined by the volume of the container used to hold the sample; this volume includes pore space within and between the sample particles inside the container [66].

### 3.2.4. Surface Area

The biochar's surface area is the interface where various biological and chemical activities take place [67]. It is closely related to the release of volatile gases during pyrolysis. A large surface area and volume of macropores offer increased water retention and gas absorption capacity. High pyrolysis pressure can increase the specific surface area [68].

### 3.2.5. Water Holding Capacity

Surface functional groups are directly related to hydrophobicity, whereas water holding capacity depends mainly on the porosity of the biochar's bulk volume. Increasing pyrolysis temperature results in a more hydrophobic biochar, holding greater amounts of water in the available pores of biochar [65].

### 3.2.6. Ash Content

The ash content (AC) of biowaste has a direct impact on the ash content of the produced biochar. During the thermochemical process of biowaste, water and volatile matter are released and result in the ash formation. Increasing pyrolysis temperature leads to increased ash content (shown in Table 3) [65].

### 3.2.7. Electrical Conductivity

The electrical conductivity (EC) of biochar is likely related to ash content. The EC of biochar increases with increasing pyrolysis temperature and residence time [67].

### 3.2.8. Cation Exchange Capacity

Cation exchange capacity (CEC) is the measure of surface charge on biochar. High CEC indicates the ability of biochar to absorb nutrients [52]. CEC reductions are due to the oxidation of aromatic C, favored at high pyrolysis temperature, forming carboxylic groups [69]. CEC increases with the age of biochar due to the increase of functional groups on its surface (shown in Table 4) [70].

### 3.2.9. Fixed Carbon and Volatile Matter

After the thermochemical process of the produced biochar, as volatile components are released, the remaining carbon content is known as fixed carbon. Biochar as a soil improver needs a significant amount of fixed carbon content. Slow heating rates and low levels of pyrolysis temperature have a direct impact on the available fixed carbon and volatile matter (Table 3).

### 3.2.10. Elemental Composition

Carbon is the basis of biochar [71]. The total carbon content of the various biochar increases with increasing pyrolysis temperature (Table 4) due to the disintegration of bonds within the biochar, causing a decrease in hydrogen and oxygen amount [64]. Hydrogen is a structural component of biochar and plays an important role in the presence of ionized molecules inside it. Oxygen is present in inorganic and organic phases of biochar. Nitrogen content is higher in biochar produced from agricultural biomass because it is present in feedstock in the form of amino acids, proteins, and pyridine. Hydrogen, oxygen, and nitrogen content decreases with increasing pyrolysis temperature.

### 3.2.11. Metals Content

The composition of metals (some of which might be heavy) in biochar depends on the composition of feedstock and pyrolysis temperature. In low pyrolysis temperature, an increase in the concentration of metals is observed, because metals are not easily evaporated, resulting in their accumulation in the solid biochar [72].

### *3.3. Factors Affecting Biochar's Quality*

Pyrolysis conditions such as temperature, pressure, residence time, heating rate, and particle size are the main parameters controlling the yield and quality of biochar, besides feedstock.

### 3.3.1. Pyrolysis Temperature

Pyrolysis temperature is a vital thermodynamic parameter of the process that affects the structure and properties of biochar. The surface area, ash content, cation exchange capacity, and pH of biochar increase with increasing pyrolysis temperature, while its yield decreases [73] due to the separation of volatile compounds (shown in Tables 3 and 4), resulting in a higher number of pores and a larger surface area [74], therefore being an excellent material for metal ions' absorption [3].

### 3.3.2. Pyrolysis Heating rate and Residence Time

A low pyrolysis temperature and heating rates favor biochar formation, while a high temperature and heating rate favor the release of gaseous products. Increased residence time leads also to an increase in biochar surface area [75,76].

### 3.3.3. Feedstock Type

The biochar's nutrient content depends on the type of pyrolysis feedstock. The contents of N and P are usually higher in biochar produced by animal manure. The order is: animal waste > agricultural waste > lignocellulosic biowaste > sludge. C is usually higher in biochar produced by lignocellulosic biomass than those produced from agricultural and animal residues because the lignocellulosic wastes are richer in C than manures [77]. O and N concentration is lower when raw materials are rich in mineral materials, such as animal manure [78], because nutrients that are more volatile, such as N, are almost transferred in gaseous pyrolysis products and less in biochar [79]. P is found to be higher in biochar derived from animal waste, because animal waste generally contains high levels of P compared other biogenic waste and sewage sludge.

Table 3, as well as Table 4 show the impact of pyrolysis conditions and the type of feedstock on the main physical and chemical properties of biochar produced by:

- Agricultural waste
- Lignocellulosic waste
- Animal waste
- Sludge

Figure 3 shows the yield of biochar produced by the four categories of Mediterranean biowaste at temperature of 300–600 °C. It shows the effect of both pyrolysis temperature and type of feedstock on the produced biochar' yield. The yields of biochar obtained at 300 °C from sewage sludge are the highest. The order is: sewage sludge > animal waste (poultry litter) > agricultural residues (wheat straw) > lignocellulosic wastes (oak tree). The same order is observed for all pyrolysis temperatures, which is clearly demonstrated in the multiparametric Figure 3.

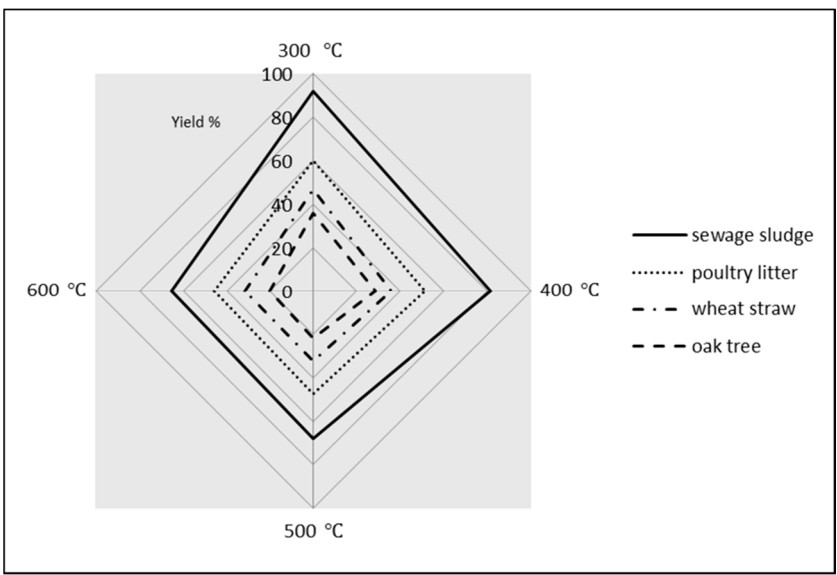

**Figure 3.** Pyrolysis temperature and feedstock effect on biochar's yield.

Figure 4 shows the effect of pyrolysis temperature and type of feedstock on biochar surface area. Selecting the appropriate data from Table 3 for particular feedstocks such as sewage sludge, poultry litter, wheat straw, and oak tree (from the four main categories) produced at a temperature of 300–600 °C, the multiparametric diagram shows that biochar' surface area is changeable. Increasing pyrolysis temperature results in increased biochar surface area. At a pyrolysis temperature of 300 °C, where the biochar's yield is the best, the surface areas of biochar from all four categories of biowaste biochar are low, with the highest being that of lignocellulosic biomass. At the temperature of 600 °C, the order for the surface area is: lignocellulosic waste (oak tree) > sewage sludge > animal waste > agricultural

waste. At all pyrolysis temperatures, animal and agricultural waste derived biochar show negligible surface area.

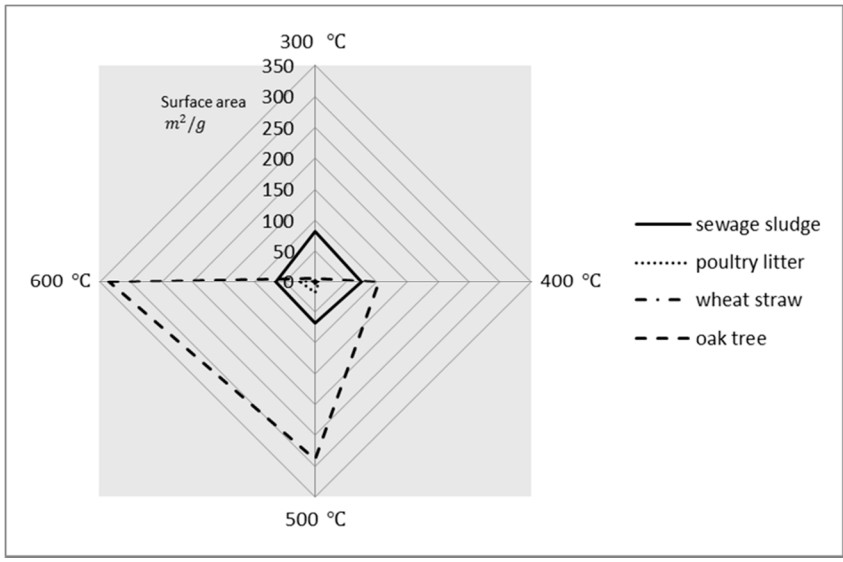

**Figure 4.** Pyrolysis temperature and feedstock effect on biochar's surface area.

The same procedure is followed in Figure 5, showing the effect of pyrolysis temperature and feedstock on biochar. Selecting the same main types of biochar (Table 4), it is observed that the pH value increases with increasing pyrolysis temperature. At a pyrolysis temperature of 300 °C, where the biochar's yield is the best, wheat straw derived biochar has the highest pH. The order is: agricultural waste > sewage sludge > lignocellulosic waste > animal waste. The biggest pH values are attributed to wheat straw derived biochar at 600 °C, with a significant difference from the other biochars. Animal waste (poultry litter), sewage sludge, and lignocellulosic (oak tree) waste give almost the same biochar pH at 600 °C. This is observed for all pyrolysis temperatures.

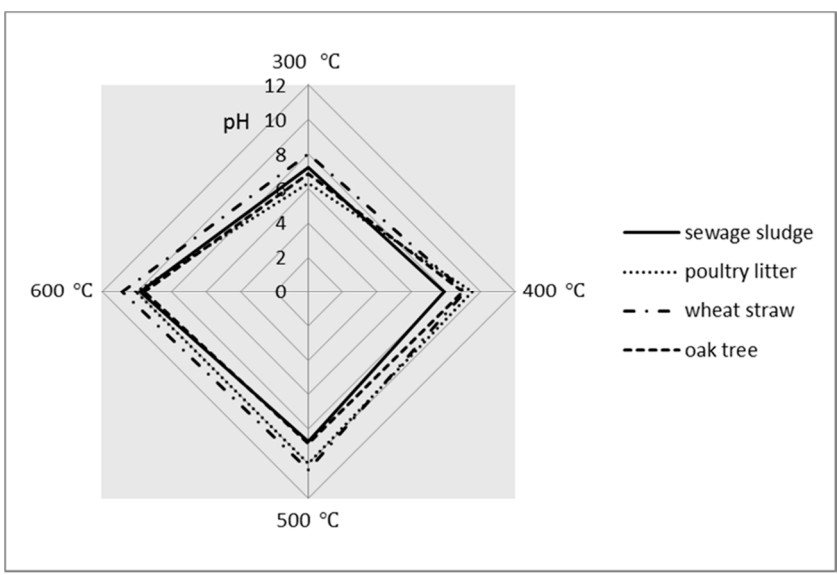

**Figure 5.** Pyrolysis temperature (300–600 °C) versus biochar pH values.

For the same types of biochar, Figure 6 shows the effect of feedstock from all four categories on biochar's C content (wt.%) at various pyrolysis temperatures (300–600 °C). At 300 °C, the order is: agricultural waste (wheat straw) > lignocellulosic waste (oak tree) > animal waste > sewage sludge.

However, increasing pyrolysis temperature results into an increase of C content in biochar. At pyrolysis temperatures of 500 and 600 °C, the order is: lignocellulosic waste > agricultural waste > animal waste > sewage sludge.

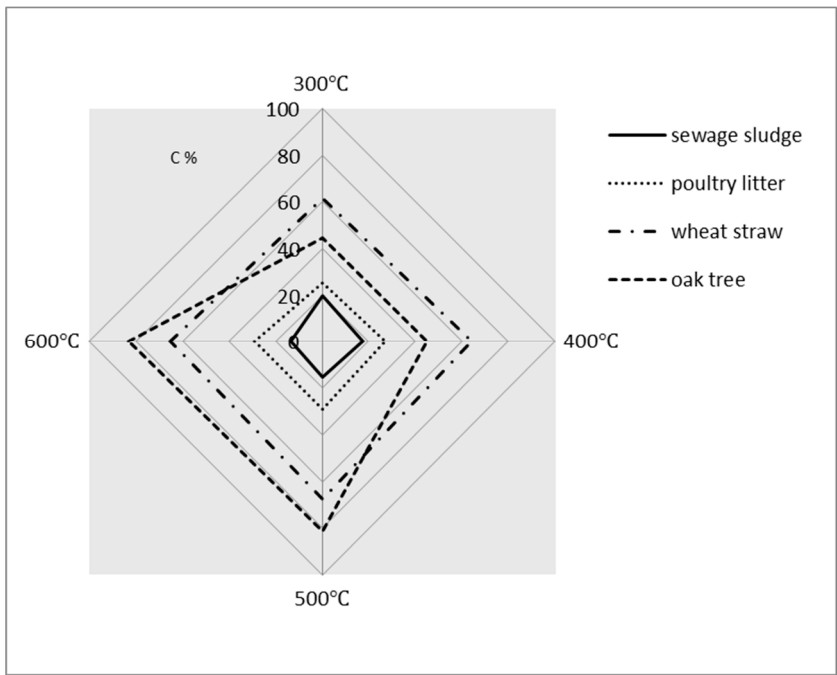

**Figure 6.** Pyrolysis temperature and waste type versus biochar C content.

For each of the above figures, representative data of the four categories were imported into Excel software and edited by using radar charts. A radar chart is a graphical method of displaying multivariate data in the form of a three-dimensional chart of three quantitative or non-quantitative variables represented on axes starting from the same point. (https://www.exceltip.com/excel-chart/radar-chart-in-microsoft-excel-2010.html).

The variables used in radar charts were:

(a)  pyrolysis temperature,
(b)  type of feedstock,
(c)  the related biochar property that is studied.

In radar charts, the alteration of the examined variable (biochar's property) is displayed, showing the significance of this alteration. Each radar chart is explained.

### 3.4. Long-Term Field Experiments in Soils with Biochar

Recommendations given by researchers on biochar applications need to be translated into the pilot scale and commercial level, under normal environmental conditions. Peer-reviewed papers discuss biochar large-scale field trials in many places, using different pyrolytic biochar for assessing crop yield with positive results in most of the cases [80]. Field experiments were reported in the literature of the last ten years to last for some seasons to less than two years. Many studies focused on relatively short periods of time. Many field studies exploring biochar' effects on plant productivity and soil quality were limited to just one or two seasons, particularly in temperate agroecosystems [81]. A field experiment on the enhancement of crop yield by rice straw and corn stalk derived biochar in Northern China lasted one year [82]. There were other studies that lasted < 2 years. Another study was carried out as a field experiment over a period of 18 months to explore the impact of co-application of biochar with sewage sludge to soil on toxicity [83].

However, longer term studies (>five years) of treatments to enhance soil aggregation, such as the addition of biochar with labile carbon to derive microbial binding agents, are limited, especially in temperate climate [84]. The literature showed that field experiments need more time due to the aging process of biochar, which undergoes different changes with a significant impact not only on soil properties but also on life [85]. C mineralization and microbial activity in biochar's field experiments found that they need several years after incorporation [86].

Only for the years 2019–2020, we found studies to refer to field experiments that lasted >5 years. This might be due to long-lasting research funding support from the EU or other sources. Long-term field trials (>5 years) are needed to explore biochar behavior in soil [87]. Some other researchers found that the soil organic carbon and total nitrogen increased after an eight year field study [88] and that biochar altered soil organic carbon and SOC based on an eight years field experiments [89], while the influence of biochar added to an agricultural soil on polycyclic aromatic hydrocarbon (PAH) levels, PAH diagnostic ratios, and soil properties was investigated, in a five year field experiment. Even after five years of experiments, the original PAH levels were not restored; therefore, more time than five years was needed [90]. The greatest potential benefit of biochar returns on bacterial community structure among three maize-straw products obtained after an eight-year field experiments in Mollisols, was recently published [91].

In addition, applied rates of biochar are correlated to the soil bulk density and pH and soil types, and these needs years of field investigations and some years of experimentations [92,93].

A recent review study suggested the "biochar carbon added" parameter as a robust comparison of different biochar-soil studies [94].

Results from large-scale biochar applications on various soil types are summarized in Table 5.

### 3.4.1. Biochar Enhances Crop Production

Many peer-reviewed papers discuss large-scale biochar field trials in many places, using different pyrolytic biochar for assessing crop yield with positive results in most of the cases [80].

The potential benefits of biochar application in soils are:

- Improved water retention capacity, leading to lower watering requirements [95].
- Improved nutrient retention [96].
- Reduced leaching of nitrogen into ground water [80].
- Removal of cations from the soil, such as heavy metals [97].
- Increased cation-exchange capacity, resulting in improved soil fertility [80].
- Removal of organic substances, such as hydrocarbons and pharmaceutical materials [98].
- Crop yield enhancement [99].
- Improved soil structure and pH value, effectiveness of fertilizer use, and reduction of toxicity [3].

### 3.4.2. Effects of Biochar on the Physicochemical and Biological Soil Properties

Biochar enhances plant growth by changing the soil structure, causing a clear increase in surface area, aeration of soil, porosity, water retention, and nutrients [73]. It stimulates the activity of a variety of microorganisms, greatly affecting the microbiological properties of soils, providing its pores as a habitat for many microorganisms, protecting them from predation and drying. Biochar essentially reduces the leaching of soil nutrients, while enhancing the availability of nutrients for plants, and reduces the bioavailability of heavy metals [73], affecting soil ecology [80].

### 3.4.3. Effect of Biochar on Soil's pH and Nutrient Content

pH and nutrient content, which are chemical properties of the soil responsible for plant growth, are balanced by biochar addition. Low pH in soils can lead to plant toxicity due to available metals and toxic ingredients, significantly increasing soil pH and thus reducing the toxic effects of the contaminants [33]. Additionally, the availability of basic micro- and macro-nutrients is significantly

increased in the soil, resulting in greater availability of primary and secondary nutrients such as K, P, Ca, and Mg [100].

Figure 7 shows the effect of biochar produced from the four Mediterranean categories of wastes at different pyrolysis temperature, on soil's pH, by considering the application rate of biochar (measured in Mg biochar per ha of soil = 1000 kg per ha) as the key parameter. It is depicted that the highest soil pH is achieved by applying biochar derived from animal waste (poultry manure), with an application rate of 40 Mg/ha.

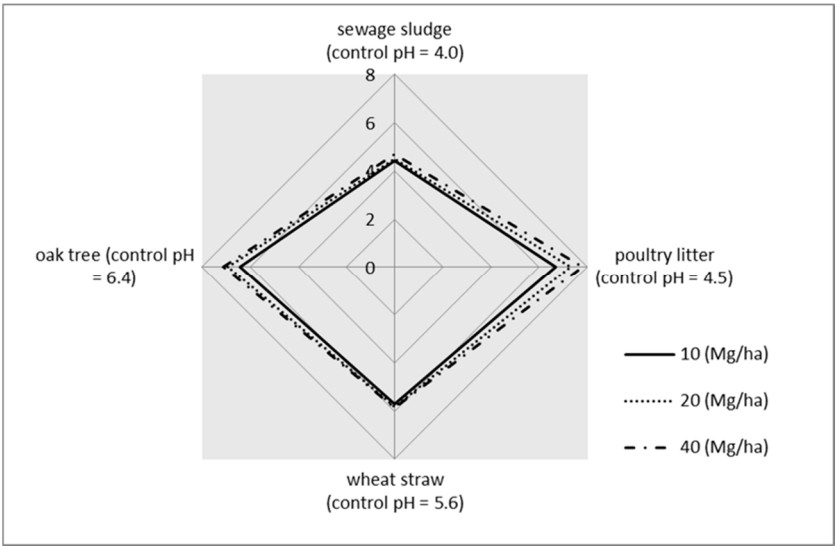

**Figure 7.** Soil pH modifications versus biochar type and application rate (Mg biochar per ha of soil = 1000 kg per ha of soil).

### 3.4.4. Effect of Biochar on CEC

Increasing pH can significantly affect CEC in many soils [33]. This is attributed to:

- The dominance of negatively charged surface functional groups.
- The increased active surface area.
- The adsorption of highly oxidized organic matter.
- The presence of residual volatile matter [100].

Table 6 summarizes the results from various biochar applications in soil, on pH and CEC, classified as per the four categories of Mediterranean wastes.

**Table 6.** Biochar field applications and impact on pH and CEC of soil values.

| Feedstock | Pyrolysis Temp (°C) | Application Rate (Mg/ha) | Soil Type | pH | | CEC (cmol/kg) | | Reference |
|---|---|---|---|---|---|---|---|---|
| | | | | Control | Treatment | Control | Treatment | |
| **Agricultural residues** | | | | | | | | |
| Green waste | 450 | 10<br>50<br>100 | Alfisol | 4.5 | 4.75<br>5.38<br>5.99 | 4.03<br>-<br>- | 10.5<br>-<br>- | [39] |
| Wheat straw | 350-550 | 10<br>20<br>40 | Anthrosols | 5.6 | 5.70<br>5.81<br>5.86 | | | [101] |
| Wheat straw | 350-550 | 10<br>20<br>40 | Halpudept | 6.5 | 6.75<br>6.77<br>6.77 | | | [45] |

**Table 6.** *Cont.*

| Feedstock | Pyrolysis Temp (°C) | Application Rate (Mg/ha) | Soil Type | | pH | | CEC (cmol/kg) | Reference |
|---|---|---|---|---|---|---|---|---|
| **Animal wastes** | | | | | | | | |
| Poultry litter | 550 | 10<br>25<br>50 | Alfisol | 4.5 | 6.66<br>7.29<br>7.78 | | | [39] |
| **Lignocellulosic waste** | | | | | | | | |
| Oak tree | | 5<br>10<br>20 | Hapludolls | 6.4 | 6.4<br>6.9<br>7.1 | 17.1 | 19.8<br>20.7<br>20.8 | [102] |
| Eucalyptus | 350 | 6<br>12<br>18 | Haplustox | 5.0 | 5.0<br>5.2<br>5.4 | 108.2<br>-<br>- | 118.5<br>131.7<br>131.5 | [55] |
| **Sludge** | | | | | | | | |
| Sludge-wood chip | 550 | 10<br>10 | Ferrosol<br>Calcarosol | 4.2<br>7.67 | 5.93<br>7.6 | 31.0<br>- | 29.3<br>- | [51] |
| Sewage sludge | 550 | 10<br>20<br>40 | Entisol | 4.0 | 4.4<br>4.5<br>4.7 | | | [103] |

### 3.4.5. Effects of Biochar on Recycling of N and P to Soils

The application of biochar increases soil's fertility by retaining metals and micronutrients in the soil. Nitrogen (N) is a necessary ingredient for crop development and is extremely exposed to losses due to its high volatility, leaching, and denitrification process. The application of biochar is shown to reduce the losses of nitrogen oxides $NO_3^-$ and N leaching.

Biochar increases crop yield because it:

- Accelerates nitration [104].
- Affects denitrification [105]
- Reduces ammonia evaporation [106].
- Accelerates soil's N transformations by increasing pure N mineralization [10,107].
- Through the adsorption of ammonia, N can be stored in soils [64].
- Biochar with its high ion exchange capacity can alter the availability of P, enhancing the ability to exchange ions or affecting the action of cations interacting with P [69].

### 3.5. Biochar's Role in Climate Change Mitigation

In Mediterranean areas, there is increasing concern about climate change impacts on agricultural production and climate change-based hazards. Increased unsustainable agricultural practices are the major contributors of greenhouse gas emissions (GHG) to the environment. Agricultural activities contribute to ~30% to the total anthropogenic emissions [108]. The main source of anthropogenic emissions of $N_2O$ is agriculture, mostly when manure and slurry are applied to fields [109].

Climate change mitigation requires carbon sequestration in addition to greenhouse gas emission reductions. Agriculture may have a high potential for carbon sequestration due to improved practices [110]. Therefore, there is a need to establish effective agricultural management practices that can mitigate GHG emissions while increasing crop production. The application of biochar to agriculture has been proposed as an appealing approach for mitigating GHG emissions and improving crop productivity [111]. Application of biochar in agriculture is an appealing approach for mitigating GHG emissions in parallel with the improvement of crops' productivity [3] and energy production. Pyrolysis for biochar is proposed as a technology for climate change mitigation.

The use of agricultural wastes and other wastes via pyrolysis offers climate change benefits, such as:

- Avoiding agricultural waste' burning and therefore reduction of $CO_2$ emissions
- Avoiding landfilling of other wastes, resulting in reducing GHG emissions.

- Offering carbon (C) sequestration by capturing and storing C in soil, preventing releases into the atmosphere [73].
- Accelerating the decomposition of soil organic carbon (SOC) [108].
- Reduction of $CO_2$ emissions; pyrolysis does not release $CO_2$ into the atmosphere as combustion does.
- Reduction of $N_2O$ emissions, which is explained by the physical or biological immobilization of $NO_3^-$ Biochar with lower N content was found to be more suitable for mitigation of $N_2O$ emissions from soil [73].
- Reduction of $CH_4$ emissions that are produced by soil microorganisms under anaerobic conditions through methanogenesis [73].
- Closed carbon cycle: By using plant residues in biochar production via pyrolysis, the carbon cycle is closed due to photosynthesis and plant growth [109].
- Biochar systems are at least GHG-neutral and/or negative. They are used to draw down atmospheric carbon by increasing stable soil carbon levels, alleviating GHG emissions because they are produced via pyrolysis, which results in an offset of fossil fuel use through simultaneous bioenergy production [110].
- Biochar systems return to the soil the nutrients from various food supply chains and mitigate climate change, by sequestrating carbon (C) [111].

Biochar production is also an important option for bioregions' circular economy and an effective tool for bioenergy conversion and as an atmospheric carbon sink to counteract climate change [112].

Finally, the most important contribution of biochar to climate change lies mainly in its role in carbon sequestration, as is explained below.

### 3.5.1. Biochar for Carbon Sequestration

Biochar production has been proposed as a technology for climate change mitigation (IPCC 2014). Biochar is carbon negative, which can reduce greenhouse gas emissions (GHGs) via carbon sequestration, with net carbon withdrawal of about 20% from the atmosphere [102,112]. Carbon sequestration is a process in which carbon is captured and stored to prevent it from being released into the atmosphere, reducing carbon emissions in the atmosphere [3,113]. The global potential for annual sequestration of atmospheric $CO_2$ through biochar application has been estimated at the billion-ton scale in Gt/year [102]. With the application of biochar in soils, the de-organization of soil organic matter happens, which is found to be higher in low-fertility soils than in high-fertility soils [113].

The addition of biochar to soils may accelerate or de-accelerate the decomposition of soil organic carbon (SOC). The acceleration of SOC decomposition might be due to:

(i) Increase in the pH to a near-neutral condition by the ash of biochar.
(ii) Improved soil moisture retention through biochar water retention.
(iii) Increase in soil aeration.
(iv) The presence of organic nutrient compounds in biochar [114].

The de-acceleration of SOC decomposition may be due to:

(i) The formation of soil aggregates, as biochar may enhance the formation of microaggregates that physically protect the SOC against decomposition [115].
(ii) The toxicity of biochar.
(iii) The sorption of enzymes and SOC to the biochar surface.
(iv) Preferential utilization of biochar rather than SOC by the microorganisms [114].

These mechanisms result in a change in the dynamics of C decomposition in soils amended with biochar.

A review study revealed some quantified indicators of biochar sequestration capacity. The characteristics of biochar produced at a higher pyrolysis temperature were: ratio of O/C < 0.2, ratio of H/C-organic < 0.4, and volatile matter < 80%; and due to these characteristics, it may have high carbon sequestration potential [116].

3.5.2. Biochar Impact on Greenhouse Gas Emissions

Climate change is transforming the planet's ecosystems and threatening the well-being of current and future generations. To keep the increase of global temperature below 1.5 °C, deep cuts in global emissions are urgently required. Greenhouse gas emissions (GHGs) are carbon dioxide ($CO_2$), methane ($CH_4$), and nitrous oxide ($N_2O$), the main contributors to deteriorating air quality, produced from fossil fuel combustion, industrial procedures, and agronomic practices.

$CO_2$ emission reductions that limit global warming to 1.5 °C can involve different mitigation measures and face different implementation challenges and potential synergies and trade-offs with sustainable development. The IPCC Special Report was on the impacts of global warming of 1.5 °C above pre-industrial levels and related global GHG pathways, in the context of strengthening the global response to the threat of climate change, sustainable development, and efforts to eradicate poverty [117].

Biochar as the product of a low pyrolysis process can be used as a tool for sequestering carbon in soil to offset greenhouse gas (GHG) emissions. However, there is a knowledge gap on the mechanisms responsible for GHG emissions [93].

The main role of biochar systems as mitigators of climate change is the increase of the stability of the organic matter. This stability is achieved by the conversion of organic materials, which mineralize comparatively quickly, into biochar, which mineralizes much more slowly [118].

The application of biochar in soil has a significant impact on GHG emissions' reduction mainly since it is a product of pyrolysis. Pyrolysis to biochar and bioenergy systems have the potential to mitigate GHG emissions through various pathways [119]:

(a)　By preventing GHG emissions from the combustion of biowastes and landfilling.
(b)　The environmental challenges caused by agricultural and animal waste and sludge disposal in the Mediterranean can be reduced by recycling these wastes via pyrolysis into biochar and energy.
(c)　Livestock manure, along with waste residues and sludge materials, precursors of biochar, emit significant amounts of GHGs, adding to global warming and deteriorating air quality.

The main advantages of pyrolysis-to-biochar systems are:

i.　Converting photosynthetic biomass carbon (C) into biochar, closing the $CO_2$ cycle due to the photosynthesis reaction.
ii.　Pyrolysis increases the recalcitrance of organic materials and enhances their activities as physical, chemical, and biological soil conditioners [120].
iii.　Pyrolytic biochar, by replacing manure and slurry application to the fields that are the main source of anthropogenic emissions of $N_2O$, contributes to the decrease of $N_2O$ release.
iv.　Biochar can act as a sorbent for organic and inorganic contaminants and can efficiently remove these materials from soils.
v.　Biochar can help improving food security by contributing to sustainable agriculture.
vi.　Biochar amendments enhance soil quality, increasing biomass production.
vii.　Soil biochar applications may directly reduce GHG emissions from soils.

However, soil biochar applications' direct effect on GHG emissions is a complex phenomenon depending on the soil's biogeochemical processes and the interactions of biochar with soil types. Soil temperature and moisture also play a role; however, these relationships are poorly understood. Researchers who performed field-scale experiments showed that biochar had no significant effect on cumulative soil $CO_2$ emissions, but they did reduce $N_2O$ emissions [120].

Researchers have compiled data from individual experimental studies and tried to quantify the effect of soil biochar applications on GHG flows ($CO_2$, $CH_4$, and $N_2O$). They discovered that biochar application significantly increased soil $CO_2$ flows by 22.14%, but also decreased $N_2O$ flows by 30.92% and did not affect $CH_4$ fluxes. They have concluded that biochar application may significantly impact global warming potential (GWP) of total soil GHG flows due to the large stimulation of $CO_2$ flows. However, soil GHG fluxes mainly varied with biochar feedstock type and soil texture, and the pyrolysis temperature, soil and biochar pH, biochar applied rate and latitude also influenced soil GHG fluxes [121].

The $CO_2$ adsorption capacity of biochar depends mainly on its physicochemical properties, such as surface area, porous structure and volume, alkalinity, inorganic composition, the presence of surface functional groups, hydrophobicity, and non-polarity. Biochar's $CO_2$ absorption capacity can be enhanced by increasing the alkalinity of the biochar surface [73]. Researchers found that by adding biochar into soils, GHG emissions, especially $CO_2$, significantly reduced compared to non-pyrolyzed materials, confirming the importance of feedstock choice for biochar production, with recalcitrance being an important initial characteristic [120].

The main mechanisms that contribute to $N_2O$ formation from unamended soils are:

(i)    Nitrification.
(ii)   Denitrification.

These pathways are related to soil physical properties such as moisture content and aeration. A large scale of soil pH and higher soil aeration may have no impact on $N_2O$ emission from biochar-amended soils. Reduction in $N_2O$ emission is explained by physical or biological immobilization of $NO_3^-$ Biochars with lower N content were found to be more suitable for mitigation of $N_2O$ emissions from soil [73].

A review study revealed that biochar with a lower N content, and consequently a higher C/N ratio (>30), are more suitable for mitigation of $N_2O$ emissions from soils [121].

Little is known about biochar interactive effect on $CH_4$ emissions and the underlying microbial mechanisms. $CH_4$ is produced by soil microorganisms under anaerobic conditions through methanogenesis. It appears that the amounts of $CH_4$ emitted depend on the physical and chemical properties of the biochar, the type of soil, as well as the soil microorganisms, and water and fertilizer management [73]. Results of experimental studies suggested that biochar soil applications could significantly mitigate the $CH_4$ and $N_2O$ emission risks under a straw return practice, but via regulating functional microbes and soil physicochemical properties; the performance of this practice always depends on soil material characteristics [122].

Finally, there is limited information on the simultaneous effects of biochar amendments on soil GHG fluxes and their global warming potential (GWP) [123].

## 4. Biochar Classification System and Associated Test Methods

The negative implications and harmful effects on the ecological system due to the continued use of biochar have not been wholly understood [124]. However, sustainable biochar industries need to provide certainty to consumers and markets, as well as safe biochar application as a soil amendment for the safety of food supply. Biochar standards provide requirements for biochar that will aid researchers to link specific functions of biochar to its beneficial soil and crop impacts. The accumulation of heavy metals, particularly arsenic, cadmium, chromium, copper, lead, nickel, selenium, and zinc, is of great concern in agriculture due to the potential threat to human and animal health. The composition of metals (heavy or not) in biochar depends on the composition of the feedstock and pyrolysis temperature [72]. This is one of the principal reasons for existing limitations on sludge derived biochar's use in soils.

However, identification of the sources of hazards at the initial stage of the biochar production process, by not using contaminated feedstock/biowastes, along with pyrolysis optimization towards

engineered biochar suitable for crops growth, can prevent risks and hazards to human health and the environment. The primordial stage of biochar supply chain assessment is at the stage of feedstock suitability. This requires thus an extensive analysis and comprehensive data on biowaste and every type of organic material if they are intended to be potential precursors of biochar [125].

At the stage of biochar commercialization, a standardization process is needed before biochar enters the markets. There are two systems of biochar products' standardization:

(a)　The International Biochar Initiative (IBI).
(b)　The European Biochar Certificate (EBC).

The classification systems aim to enable stakeholders and the market to identify the most suitable biochar to fulfil the requirements for soil and/or land use [126].

Towards circular economy strategies, the use of biochar as soil amendment and for climate change mitigation provides circular economy options. For this reason, the EU and national legislations must adequately prepare to regulate both the production and the application of biochar and to integrate it in the circular economy proposals. However, countries are behind in establishing national regulations of biochar in accordance with waste management and fertilizer directives [127].

Biochar quality standards have been formed in Europe with the European Biochar Certificate (EBC), in the U.K. with the Biochar Quality Mandate (BQM), and in the USA with the IBI standard, which is intended to be used internationally, for filling the need for biochar regulations. In parallel with this, biochar producers and biochar users in EU countries were partly successful in fitting the new biochar product into the existing national legislation for fertilizers, soil improvers, and composts. The EBC and IBI Guidelines for Biochar were developed independently since autumn 2009 (IBI), respectively spring 2010 (EBC). The main differences lie in the fact that the EBC integrates an on-site control of sustainable production, whereas the IBI is based on a voluntary testing of any produced biochar. A comparison of the EBC standards with IBI Standards appeared first on-line in 2017.

In this study, we re-summarize this comparison in Table 7, with the scope to contribute to the awareness of required standardization and testing needs.

**Table 7.** Comparison of European Biochar Certificate (EBC) standards (Version 4.8) with International Biochar Initiative (IBI) standards (Version 2.0).

| Parameter | EBC V4.8 | EBC Test Method | IBI V2.0 | IBI Test Method |
|---|---|---|---|---|
| **Physical Properties** | | | | |
| **Bulk density** | Required | DIN 51705 | Not Required | N/A |
| **Particle size distribution** | Not Required | N/A | Required | Progressive dry sieving with 50, 25, 16, 4, 2, 1, 0.5 sieves |
| **Water content** | Required | DIN 51718 | Required | ASTM D1762-84 |
| **Surface area** | Required | Milled < 50μm, 2 h at 150 °C vacuum, $N_2$ | Optional | ASTM D6556 |
| **Water holding capacity** | Optional | E DIN ISO 14238 | Not Required | N/A |
| **Chemical properties** | | | | |
| **Electrical conductivity** | Required | DIN ISO 11265 | Required | U.S. Composting Council |
| **Total ash** | Required | DIN51719, ISO1171, EN14775 | Required | ASTM D1762-84 |
| **pH** | Required | DIN ISO 10390 | Required | U.S. Composting Council |
| **Total C** | Required | DIN 51732, ISO 29541) | Required | ASTM D4373 |
| **Molar H/C$_{org}$ ratio** | Required | DIN 51732, ISO 29541) | Required | ASTM D4373 |
| **Molar O/C ratio** | Required | DIN 51732, ISO 17247 | Not Required | N/A |
| **N, P, K Content** | Required | DIN 51732, ISO 29541 | Required | ASTM D4373 |

**Table 7.** *Cont.*

| Parameter | EBC V4.8 | EBC Test Method | IBI V2.0 | IBI Test Method |
|---|---|---|---|---|
| **Volatile matter VOCs** | Optional | TGA 701 (Thermogravimetric Analysis) | Optional | ASTM D1762-84 |
| **PAHs** | Required | DIN EN 15527, DIN CEN/TS 16181 (European Standarisation) | Required | U.S. EPA 8270 |
| **Pb, Cd, Cu, Ni, Hg, Zn, Cr Content** | Required Basic grade: Pb < 150 mg/kg Cd < 1.5 mg/kg Cu < 100 mg/kg Ni < 50 mg/kg Zn < 400 mg/kg Cr < 90 mg/kg Premium grade: Pb < 120 mg/kg Cd < 1 mg/kg Cu < 100 mg/kg Ni < 30 mg/kg Hg < 1 mg/kg Zn < 400 mg/kg Cr < 80 mg/kg | All metals: DIN EN ISO 17294-2Hg: DIN EN 1483 | Required As: 12–100 mg/kg Cd: 1.4–39 mg/kg Cr: 64–1200 mg/kg Co: 40–150 mg/kg Cu: 63–1500 mg/kg Pb: 70–500 mg/kg Hg: 1–17 mg/kg Mo: 5–20 mg/kg Ni: 47–600 mg/kg Se: 2–36 mg/kg Zn: 200–7000 mg/kg B Declaration Cl Declaration Na Declaration | All elements except Hg and Cl: i. Microwave-assisted $HNO_3$ digestion ii. $HNO_3$ digestion determination with iii. ICP-AES iv. Flame (according to U.S. Composting Council Sections 04.05 and 04.06) Hg: U.S. EPA 7471 Cl: Ion chromatography or ion-selective electrode |

## 4.1. International Biochar Initiative

The IBI standards are the result of an ongoing multi-year development process that is global, transparent, and non-exclusive and involves hundreds of researchers, entrepreneurs, farmers, and other stakeholders (https://biochar-international.org/).

## 4.2. European Biochar Certificate

European Biochar Certificate (EBC), first published in 2012, ensures sustainable biochar production without potential risks to agronomic systems. Based on the latest scientific data, it is economically viable and as close as possible to agricultural systems [128]. Biochar produced according to EBC standards meets all the requirements of sustainable production with at the same time a positive carbon footprint. These standards guarantee ecologically viable supplies and raw material for biochar production, in line with emission standards combining environmentally-safe storage. EBC was developed to reduce the risks of using biochar based on the best scientific knowledge and to help biochar users and producers prevent or at least reduce the risks to health and the environment during biochar production and use (http://www.european-biochar.org/en).

## 5. SWOT Analysis

SWOT analysis plays the role of the conclusions on the positive and negative aspects of biochar applications in this review. SWOT analysis is a strategic planning technique used to identify strengths, weaknesses, opportunities, and threats related to biochar application in soil. Using SWOT analysis generates meaningful information for each category to make the tool useful and identify a competitive advantage. All the internal factors below are a combination of both the reviews' outcomes and the authors' conception. The assessment of biochar is presented with the estimation of the strengths, weaknesses, opportunities, and threats in Table 8.

**Table 8.** SWOT analysis of biochar.

| Strengths | Weaknesses |
|---|---|
| √ Biochar derived from wastes is an inexpensive, sustainable, and easily-produced material with potentially extensive applications [124]. | √ Logistics and storage of waste problems and cost [129–131] |
| √ Biochar for soil application is an important means for establishing a long-term carbon sink with low-risk return of $CO_2$ to the atmosphere and the improvement in soil properties [3,124] | √ Biochar lower quality criteria |
| | √ Market based economic conditions in conservation agriculture. |
| √ Biochar increases soil fertility [3], improves soil nutrient availability and water holding capacity, hence improving degraded soils and promoting soil health and crop productivity [99] | √ Biochar with high pH is not suitable for alkaline soils [99] |
| | √ Increases C/N ratio [116] |
| | √ PAHs as potential toxic elements and pollutants [3,126] |
| √ Offers waste management options in the circular economy [129–131] | √ Biochar made from waste material streams must be under Waste Framework Directive/End-of-Waste criteria [127,128] |
| √ Promotes sustainable agriculture practices and food safety [129–131] | |
| √ Contributes to rural development | √ Biochar law must be harmonized by the countries [127] |
| √ Offers closing loops of nutrients [33] | |
| √ and a circular economy option [25,129–131] | √ Full and transparent up-to-date information on biochar production and product quality required [126] |
| √ Offers a C sequestration option [73] and contributes to GHGs' mitigation, improving air quality and mitigating climate change [3] | √ Control and continuous follow-up |
| | √ Contaminated feedstock sources should be avoided by utilizing suitable sources [124] |
| √ Offers resource efficiency [25], contributing to the resource flow balance of rural and urban areas [25,129–131] | √ Effective implementation of risk control in biochar production, management, and sustainability mechanisms [125] driven by environmental and social standards and policy |
| | √ Control of a soil applicant for contaminants and GHG management [124,125] |
| | √ Hazard control necessitates risks to be considered along all the supply chain of biochar [126] |
| | √ Considerations linked to biochar production and the wide range of its possible applications should consider the legal regulations on waste management, use of fertilizers, and product safety [126] |

| Opportunities | Threats |
|---|---|
| √ Pyrolysis is a practice of many types of waste management [3,127] | √ Biochar's risks of high soil pH increase [99] |
| | √ Possibility of soil infection and toxicity by unsuitable biochar [15,16,99] |
| √ Pyrolysis offers parallel production of energy [3,129–131] | √ Insufficient biochar specifications/standards for health and crop safety [47] |
| √ Pyrolysis offers opportunities for tailored, small-scale, local biochar facilities for rural development [125,126] | |
| | √ Environmental, ecological, and human health safety from the use of biochar products in the open soil environment [47,124,125]. |
| √ Pyrolysis is an environmentally-friendly technology, not releasing GHGs [62] | |
| √ Biochar offers opportunities for C sequestration [73] | √ Biochar products' usage, made from sludges and wastes, not regulated under mandatory EU regulations may create hazards [125] |
| √ Circular economy models can be advanced with pyrolysis-to-biochar and energy systems comping wastes from many producers and producing biochar for many agricultural users [130,131] | √ In peri-urban/urban agriculture, biochar may counter harmful compounds like heavy metals, dioxins, and PAHs (polycyclic aromatic hydrocarbons) present in raw sewage or refuse inputs |
| √ Industrial symbiosis models by using wastes from many agro-industries can be boosted [130,131] | |

## 6. Conclusions

In this review study, we focused on the pyrolysis technology as the biochar maker, pyrolytic factors as engineering, and design parameters affecting biochar properties for nutrients' recycling and soil fertility benefiting plant growth, C sequestration, and GHG emission reduction. This was undertaken by reviewing the most recent reviews and research papers found on international scientific sites and platforms and by extracting some concluded insights in a practical manner.

The bibliographic research showed that the interest in biochar is growing in an exponential way, emerging in conjunction with the major frameworks of sustainable agriculture, climate change mitigation, waste management, fertilizer use, food security, and the circular economy. The specific scientific interest is mostly focused on biochar's ability to retain carbon, nutrients, and water in soils and its potential to reduce greenhouse gas emissions (GHGs_ such as $CO_2$, $N_2O$, $CH_4$).

The literature brought much information concerning the production and characteristics of biochar and many experimental data concerning the use of biochar as a soil amending material. Fewer studies dealt with biochar as a climate change mitigator. There is still a lack of clear knowledge of the mechanisms of the above scientific topics. The investigations of biochar effects in large-scale, open field, long-term experiments are recent. Experimental works with >5-year field application studies were mostly published in the year 2020, a fact that indicates available financial support for demonstration and R&D biochar projects, in Europe and globally. Fewer studies were devoted to the risks and hazards that biochar entails and the consequent need for standards and national legislation. However, it became clear that feedstock characteristics are primordial factors of biochar toxicity.

The bibliographic screening showed that biochar has many advantages and potentialities for crop growth and air quality enhancement. Biochar amendment to soils is cost-effective and a sustainable approach to mitigate greenhouse gas emissions, improve phytoremediation, and minimize the health risks associated with consumption of chemical fertilizers. Its importance will increase in the future, because of the fast-growing urbanization and imperative need for any kind of waste management in urban agglomerations and rural areas of the Mediterranean region and globally.

There is an urgent need for sustainable management of the biowaste and sludge flows of urban agglomerations and of the agricultural and animal wastes of rural areas in Mediterranean countries. The pyrolysis of biowaste is a friendly environmental technology suggested as waste management practice for the Mediterranean urban-rural systems. Pyrolysis offers a promising approach for managing carbon-rich wastes towards engineered biochar production and closing $CO_2$ and C loops, also offering a circular economy option. Furthermore, the nutrient fluxes and balances of high input agricultural systems of the Mediterranean region, which are characterized by a high fertilizer's use, can be better balanced with biochar's use.

Due to the direct relationship between pyrolysis parameters and the type of biowaste used as biochar precursors, an extensive feedstock and biochar characterization is needed, and biochar standardization is of paramount importance in order to reveal its sustainable and safe role for agronomical and environmental uses. Some threats due to environmental risks and biochar's potential toxicity need to be carefully considered in the very beginning of the biochar process: at the stage of feedstock selection as precursors to biochar.

Biochar converts carbon into a more stable form via slow pyrolysis. The microscopic structure can be the home of microbial colonization, which retains nutrients and sequesters more carbon from the atmosphere. Its application to soil can sequestrate carbon, adsorb inorganic and organic contaminants, and improve soil fertility and quality through increases in pH, macronutrients, and improved soil water holding capacity.

Future R&D efforts are needed to focus on the following topics:

- Clarification of the mechanisms of the processes (microbial colonization, water retention, GHG reduction, climate change mitigation).
- Trade-offs between biochar application and crop yields and food safety.

- The design of biochar though pyrolysis optimization for different soil types.
- Food security and health safety issues by biochar applications.

**Author Contributions:** A.Z.: Anastasia Zabaniotou; K.S.: Katerina Stamou. Conceptualization, A.Z.; methodology, A.Z.; validation, K.S. and A.Z.; formal analysis, K.S. and A.Z.; investigation, K.S. and A.Z.; resources; A.Z. and K.S.; data curation, K.S.; writing, original draft preparation, K.S.; writing, review and editing, A.Z.; visualization, K.S. and A.Z.; supervision, A.Z.; project administration, A.Z. All authors read and agreed to the published version of the manuscript.

**Funding:** This research received no external funding.

**Conflicts of Interest:** The authors declare no conflict of interest.

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
