# Peer review of "Balancing Waste and Nutrient Flows Between Urban Agglomerations and Rural Ecosystems: Biochar for Improving Crop Growth and Urban Air Quality in The Mediterranean Region"

_atmosphere, doi:10.3390/atmos11050539_

Round 1

Reviewer 1 Report

The article makes an interesting review of the biochar produced from Mediterranean wastes.
However, the study area is not specified. Being a large area, I wonder if all the articles are representative for
extend the conclusions to the entire Mediterranean.

It does not indicate what scientific databases were used to perform a bibliographic review.
Check:
*lines: 43 - 44 harvesting better air quality in cities by sequestrating C in soil. Clearly indicate that it is C.
*line198: ofLignocelulosic

It is not indicated where or how figures 4, 5, 6, 7, 8 are obtained.

In section 3.4.5. Results from large scale biochar application in soil, an introduction is required.

Section 4. SWOT Analysis, is very poor, it is required to delve into how these results were reached. It was an expert consultation or exclusive results of the authors' perception.

Reorganization of the conclusions is required. Nothing is indicated about the bibliographic search, the main results of the review, the current situation of the research in the field studied.

The information is extensive, but as a reader there is information that is confusing and therefore must be reorganized. Likewise, I consider the intention to present the tables and graphs must be indicated. On the other hand, there is a series of charts that cannot be understood how they were produced.

Considering the above, I suggest that the document be reviewed in depth to reassess its publication.

Reviewer 2 Report

The review “Balancing waste and nutrient flows between urban agglomerations and rural ecosystems in the 3 Mediterranean area: Biochar from up-recycled urban  waste via pyrolysis, for improving crop growth and  urban air quality” describe and list the main physico-chemical characteristics and also the most important application of several kind of Biochar in agriculture field.

The title is a bit long.

Abstract: is better to avoid abbreviations in this section

Introduction: is very brief. No complete introduction to the biochar is reported. In particular, informations about the origin of this substances are not considered. Moreover notices on recent regulation for biochar miss. Finally only the applications in open field are reported and no information on the use of biochar as matrix for growth media in nursery are considered. This field of application is important aso for the replacement of the peat.

The figure 1 appear of low impact. Probably is better to change it with a schematic flow representing the biomass recicle, its use and effects for the environment. Otherwise is better to cancel it.

In table 1 at the column “Number of articles in Sample” are reported 7652 articles by searching “biochar and soil amendment”, while in the text at line 53, 4,386 publications on the use of biochar as a soil improverare reported.  7652 results adding 4386 + 3,266 resulted in publications searching "Biochar AND Green House Gases emissions/ mitigation". Please better clarify this numbers at the bigin of the paragraph

Paragraph 3.2. Biochar Properties: no sub-paragraphs on density, ash content, pH are reported. In particular the bulk density limits (mainly for substrate) and the pH are very important parameters.

Moreover between the chemical parameters no informations about the electrical conductivity are reported; please add a section.

Most important: the CSC is listed in the physical parameter of biochar section. This parameter, despite is influenced by the kind and surface area of the materials is clearly a chemical parameter.

Line 126: the list of the heavy metal is wrong. Most of them are not heavy metal. This sub-paragraph recall the table 4. But table 4 reports parameters such as pH, O, C, H, N, pyrolis temperature and the title is: Table 4. Elemental composition and pH of produced biochar. Please check it and improve the correspondence with the text. 

Paragraph 3.3. Factors Affecting Biochar’s Quality and its sub-paragraphs: please check the numbers

Line 270 3.4.4. Effects of biochar on recycling of N,P and other metals to the soil: no effets on “other metals” are reported in the paragraph.

Reviewer 3 Report

I have hard to understand what this paper is really about. The title seems to indicate that it has to do with Mediterranean areas, but in the tex I find little reference to that. It seems to be a review of biochar, but that has been reviewed before. The methodology states that it follows a methodology to select literature, but it does not explain the methodology, and it stats that many papers were excluded, without explaining why.

Many figures and tables have little information (e.g. fig. 1) and it is hard to understand why they are there. The main type of figures used (4-8) ae hard for me to understand.

In short, I see little in this manuscript worth publishing.

Round 2

Reviewer 1 Report

It is indicated in some detail how the search is carried out, but it is not specified as the articles are selected from the different databases used. That is not defined, and as it is a review it must be clearly understood.
Which are the Mediterranean countries, continues without indicating them, despite the fact that it was indicated in the previous review.

As justifies that it is representative in analysis for the Mediterranean countries. That is, within the analysis are all the countries in question.

Author Response

REPLY TO REVIEWER’s COMMENTS

Comment 1:  It is indicated in some detail how the search is carried out, but it is
not specified as the articles are selected from the different databases
used. That is not defined, and as it is a review it must be clearly
understood.

REPLY         The following text was added, highlighted in yellow:

                    Lines 160-172:

                     The bibliographic databases used for sourcing the articles were Web of Science (2009–2019), ScienceDirect (2009-2020), Google Scholar (2009-2019), MDPI (2009-2019) and Open access publications. It is recognized that there is an extensive literature in the form of books, but it was not possible to have access to all relevant books for a systematic review, however we used some. In order to keep the number of articles reasonable and to ensure the quality of the sources, the search was further restricted to peer-reviewed articles. To keep results to a manageable number, the search was restricted to the title, abstract, and keywords of papers. Document type was limited to ‘articles’ and reviews. It was decided that the final sample would be limited to papers that had been cited. There was restriction on the year of publication, or the journals considered, for the decade of  2009-2019, because most of the scientific publishing on biochar appear after 2009.

                     During the revision of the manuscript to its updated version, 13 publications of the year 2020 were also searched and cited; however, these published articles are accepted even without citations, since they all were of the year 2020, but they are not comprised in the Figures 1 and 2.

Comment 2:  Which are the Mediterranean countries, continues without indicating
them, despite the fact that it was indicated in the previous review.
As justifies that it is representative in analysis for the Mediterranean
countries. That is, within the analysis are all the countries in question."

REPLY         We cited the European Mediterranean countries, but results can be for other Mediterranean countries non-European (Turkey, Egypt etc

                     Lines 150-152

                     Assessment of current development work and evaluation of potential opportunities for available biowaste at the European Mediterranean countries (Greece, Italy, France, Spain, Portugal, etc) but also at other non-European Mediterranean countries (Tutkey, Egypt, etc)

Reviewer 2 Report

The reviewed version of the manuscript "Balancing waste and nutrient flows between urban 2 agglomerations and rural ecosystems: Biochar for  improving crop growth and urban air quality in the  Mediterranean region" Was significantly improved and the detected criticsms were solved. The correspondence between the title and the paper content is now clear . 

Also the section in which the results review are presented is well organized. 

In my opinion the paper can be published in the Atmosphere journal.

Author Response

(The authors gave the same response as above.)
